# Facile Synthesis of Super-Microporous Titania–Alumina with Tailored Framework Properties

**DOI:** 10.3390/ma13051126

**Published:** 2020-03-03

**Authors:** Yongfeng Li, Jiaojiao Su, Guiping Li, Xiufeng Meng

**Affiliations:** 1New Energy Engineering, Shanxi Institute of Energy, Jinzhong 030600, China; lyf_a@foxmail.com (Y.L.); mgylgp@sina.com (G.L.); mgymxf@126.com (X.M.); 2College of Chemistry and Chemical Engineering, Taiyuan University of Technology, Taiyuan 030024, China

**Keywords:** super-microporous alumina–titania, different solvents, fatty alcohol polyoxyethylene ether, high BET surface areas

## Abstract

Super-microporous material (pore size 1–2 nm) can bridge the pore size gap between the zeolites (<1 nm) and the mesoporous oxides (>2 nm). A series of super-microporous titania–alumina materials has been successfully prepared via a facile one-pot evaporation-induced self-assembly (EISA) strategy by different solvents using fatty alcohol polyoxyethylene ether (AEO-7) as the template. Moreover, no extra acid or base is added in our synthesis process. When titanium isopropylate is used as the titanium source, these materials exhibit high BET surface areas (from 275 to 396 m^2^/g) and pore volumes (from 0.14 to 0.18 cm^3^/g). The sample prepared using methanol as the solvent shows the largest Brunauer–Emmett–Teller (BET) surface area of 396 m^2^/g. When tetrabutyl titanate is used as the titanium source, these materials exhibit high BET surface areas (from 282 to 396 m^2^/g) and pore volumes (from 0.13 to 0.18 cm^3^/g). The sample prepared using ethanol as the solvent shows the largest BET surface area of 396 m^2^/g.

## 1. Introduction

Porous materials have received considerable attention owing to their ability to interact with atoms, ions, molecules, and nanoparticles not only at their surfaces, but throughout the bulk of the materials [1]. Therefore, the presence of pores in nanostructured materials greatly promotes their physical and chemical properties. Among these non-siliceous oxides, TiO_2_–Al_2_O_3_ is of interest for many applications in wide fields, such as catalysis [2,3,4,5], ceramic [6,7], gas sensor [8], solar cells [9,10], and others [11]. Mixed titania–alumina oxides exhibit properties superior to those of single-metal oxides (alumina or titania). Such a mixture may broaden the range of applications available to this hybrid material. For instance, tielite (Al_2_TiO_5_) is used as a potential adsorbent in the decontamination of chemical warfare agents [12], in materials for aeronautical and automotive purposes [13], and in orthopedic and dental implants [14]. The reported alumina–titania support for molybdenum dispersion used for the hydrodeoxygenation of palmitic acid exhibited excellent catalytic performance [3]. A study describes the enhanced performance of the chemical looping combustion of methane with an Fe_2_O_3_/Al_2_O_3_/TiO_2_ oxygen carrier [15]. Syntheses of mesoporous alumina–titania systems by different preparation methods have been reported. For example, Stacy M. Morris et al. [16] prepared mesoporous alumina–titania materials over a wide range of compositions by the self-assembly of Al and Ti isopropoxides and a triblock copolymer structure-directing agent. Liu Erming et al. [17] synthesized a series of macro–mesoporous titania/alumina core–shell materials in an oil/water interface. Guo Changyou et al. [18] synthesized a mesoporous Al_2_O_3_–TiO_2_ composite oxide using solvothermal method in a benzyl alcoholeoleyl amine system. However, little work has been carried out on the preparation of super-microporous (pore size 1–2 nm) titania–alumina mixed oxides. The materials in this pore size range could bridge the pore size gap between the zeolites (<1 nm) and the mesoporous oxides (>2nm). Such materials exhibit the potential of size and shape selectivity for those organic molecules that are too large to access into the pores of microporous zeolites and zeolite-like materials [19,20].

In this research, a series of super-microporous titania–alumina materials (pore size 1–2 nm) were prepared through the evaporation-induced selfassembly (EISA) method using titanium isopropylate or tetrabutyl titanate as the Ti source and aluminum nitrate nonahydrate as the Al source. There was no acid or base addition during the whole preparation process. By varying the solvent used among methanol, ethanol, 1-butanol, isobutanol, or 1-octanol, we successfully obtained titania–alumina materials with tailored framework properties. 

## 2. Materials and Methods 

### 2.1. Chemicals 

Fatty alcohol polyoxyethylene ether AEO-7 (Mav = 575-605, RO(C_2_H_4_O)_n_H, n = 7) was purchased from BASF (China) co., LTD. Shanghai branch. Aluminum nitrate nonahydrate, anhydrous ethanol, and tetrabutyl titanate were obtained from Tianjin Chemical Reagent Co. Titanium isopropylate was purchased from Shanghai Aladdin Biological Technology Co., Ltd. All the chemicals were of analytical grade and used as received without further purification.

### 2.2. Preparation of Super-Microporous Titania–Alumina Materials

In a typical synthesis, 2.0 g of fatty alcohol polyoxyethylene ether AEO-7 was dissolved in 20 mL of waterless ethanol at room temperature. Then, 5 mmol aluminum nitrate nonahydrate and 5 mmol titanium isopropylate or tetrabutyl titanate were added. Upon rapid stirring at room temperature for least 2 h, the resulting homogeneous sol was transferred to a petri dish and underwent solvent evaporation at 45 °C for two days and at 100 °C for one day. The final solid products were heated at 400 °C for 5 h to remove the organic template and named as MTA. The as-prepared super-microporous titania–alumina samples were labeled, starting with a prefix of MTA followed by the type of solvent (M, E, B, IB, and O, which refer to methanol, ethanol, 1-butanol, isobutanol, and 1-octanol, respectively), then titanium precursors (i and b, which refer to titanium isopropoxide and tetrabutyl titanate), and finally calcination temperature. For example, MTA-M-i-400 refers to super-microporous titania–alumina prepared from titanium isopropoxide with methanol solvent calcined at 400 °C for 5 h. High-temperature treatment (550 °C and 750 °C) was carried out in air for 1 h with a temperature ramp of 10 °C/min.

### 2.3. Characterization

Powder X-ray diffraction (XRD) measurements were performed using a Shimadzu XRD-6000 diffractometer made in Japan using Ni-filtered Cu Kα (0.154 nm) radiation. N_2_ adsorption/desorption isotherms at 77 K were measured on a Quantachrome QUADRASORB SI instrument. The Brunauer–Emmett–Teller (BET) method was used to calculate the specific surface area. The microporous structure was obtained from the t-plot analysis of the adsorption branch of the isotherm. The pore size distribution was calculated using the density functional theory (DFT) method pore size model applied to the adsorption branch of the isotherm. Total pore volumes were obtained at pressure 0.95 [21,22]. Micropore volumes were obtained from the t-plot method at a pressure of 0.2–0.5. Thermogravimetric-differential scanning calorimeter (TG-DSC) analysis was conducted on a NETZSCH (STA449F3) instrument made in Germany. 

## 3. Results and Discussions

Figure 1 presents the TG-DSC pattern of the obtained precursor without removal of the template when tetrabutyl titanate and ethanol are used as the titanium source and solvent. The endothermic peak at 180 °C is attributed to the evaporation of water and ethanol in the gel. The more prominent thermal event located in the 200–400 °C temperature range is attributed partly to the remaining water included in the pores and mostly from the decomposition and oxidation of the template. 

The nitrogen adsorption–desorption isotherms and the corresponding pore size distribution curves for MTA-400 samples using different solvents and titanium sources are displayed in Figure 2 and Figure 3. The detailed textural properties are listed in Table 1. It can be seen that all samples calcinated at 400 °C exhibit a typical type I isotherm with no distinct hysteresis loop, thus indicating the presence of micropores. All these samples show narrow pore size distribution around 1–2 nm. It is also observed that there is no significant relation between porous structure type and solvent. BET surface areas does not have a linear relationship with the solvent. When ethanol is used as the solvent, MTA-E-b has the bigger BET surface area among them. The isotherm obtained using MTA-E-b, yields a surface area of 396 m^2^/g of which 377 m^2^/g in the form of micropores. The surface area of MTA-IB-b is the smallest. When titanium isopropylate is used as titanium source, MTA-M-i exhibits a BET surface area of 396 m^2^/g area and a microporous surface area of 375 m^2^/g, which are much larger than other sample. The physisorption measurements reveal the largest BET surface area when methanol is used as the solvent.

XRD results show that all samples calcined at 400 °C (not shown) are amorphous without the presence of any crystalline alumina and/or titania phases, suggesting that the extremely homogeneous super-microporous titania–alumina nanomaterials are formed. With the increase of quenching temperature, some samples begin to show crystallinity (as confirmed by wide angle powder XRD patterns shown in Figure 4) at the calcination temperature 550 °C. Several relatively weak diffraction peaks at 2θ = 25.3°, 38.5°, 48°, 55.1° and 62.6° are observed, which could be indexed as the anatase phase of titania. XRD results show that sample MTA-IB-b and MTA-M-i are crystalline, consisting only of anatase titania without any traces of crystalline alumina. When the solvent is ethanol, XRD patterns of the sample show quite broad and weak diffraction peaks, which suggests that the anatase titania crystal size is extremely small. However, the materials synthesized with other solvents are still amorphous. When the temperature is up to 750 °C (in Figure 5), all the samples are crystalline. 

## 4. Conclusions

We have successfully synthesized a series of super-microporous titania–alumina materials without adding acid or base and calcining under mild conditions. It is found that the BET surface area does not have a linear relationship with the solvent. When titanium isopropylate is used as titanium source, the BET surface area of the sample prepared using methanol as the solvent is the largest at 396 m^2^/g. When tetrabutyl titanate is used as titanium, using ethanol as the solvent presents the largest BET at 396 m^2^/g. Most importantly, this work opens a new methodology for the preparation of porous titania–alumina materials with good textural properties.

## Figures and Tables

**Figure 1 materials-13-01126-f001:**
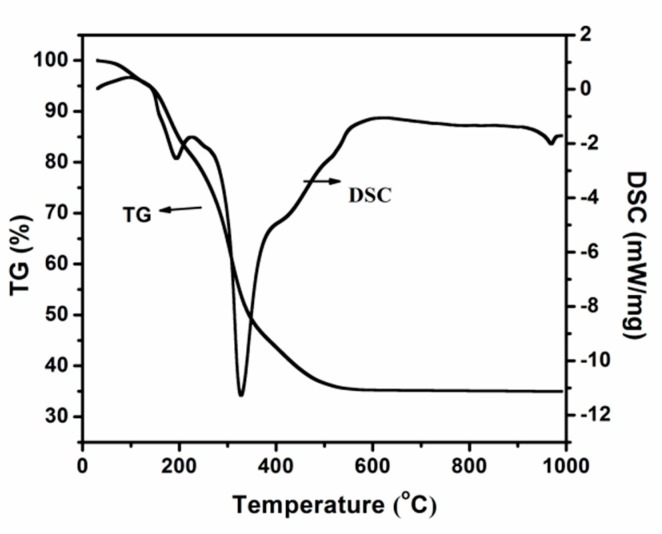
Thermogravimetric-differential scanning calorimeter (TG-DSC) curves of the obtained precursor when tetrabutyl titanate and ethanol are used as titanium source and solvent.

**Figure 2 materials-13-01126-f002:**
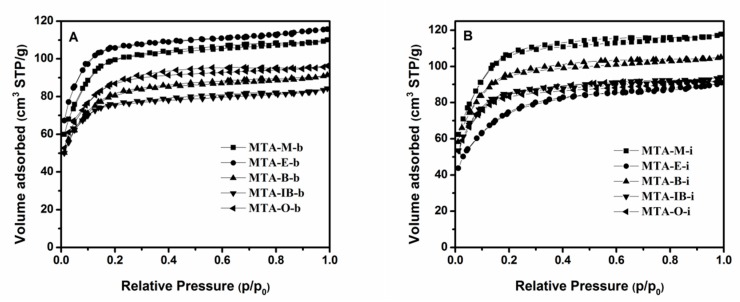
N_2_ adsorption–desorption isotherms of samples MTA-b (**A**) and MTA-i (**B**) calcined at 400 °C.

**Figure 3 materials-13-01126-f003:**
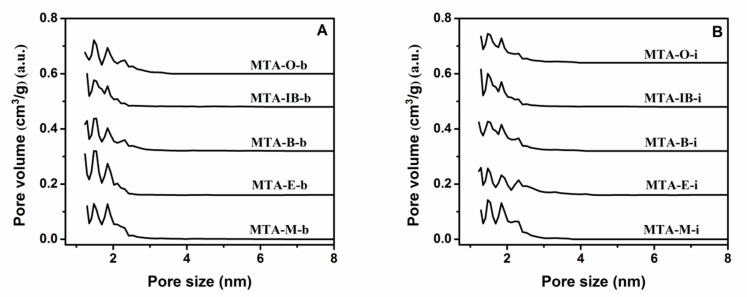
The corresponding pore size distribution curves of samples MTA-b (**A**) and MTA-i (**B**) calcined at 400 °C. For clarity, the pore size distributions curves of MTA-E-b/MTA-E-i, MTA-B-b/MTA-B-i, MTA-IB-b/MTA-IB-i, MTA-O-b/MTA-O-i are offset along the *Y*-axis by 0.16, 0.32, 0.48, and 0.64 cm^3^/g, respectively.

**Figure 4 materials-13-01126-f004:**
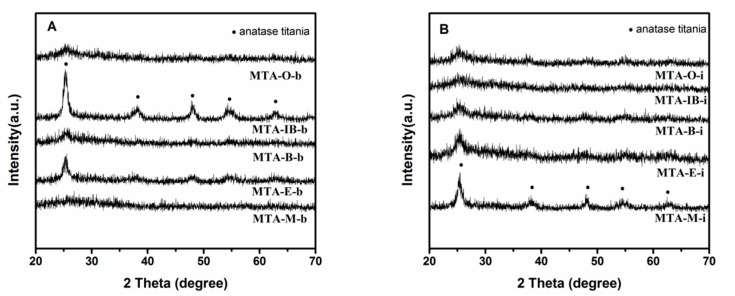
XRD patterns of samples MTA-b (**A**) and MTA-i (**B**) after calcination at 550 °C.

**Figure 5 materials-13-01126-f005:**
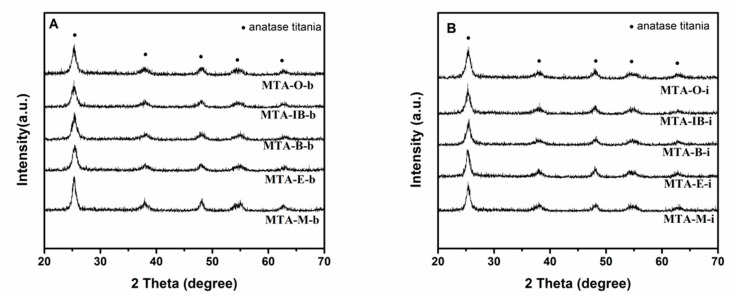
XRD patterns of samples MTA-b (**A**) and MTA-i (**B**) after calcination at 750 °C.

**Table 1 materials-13-01126-t001:** Brunauer–Emmett–Teller (BET) surface areas and pore structures of various samples calcined at 400 °C.

Samples	S_BET_ (m^2^/g)	S_mic_ (m^2^/g)	V_total_ (cm^3^/g)	V_mic_ (cm^3^/g)	Pore Size (nm)
MTA-M-b	370	350	0.17	0.15	1.5
MTA-E-b	396	377	0.18	0.16	1.5
MTA-B-b	294	277	0.14	0.12	1.5
MTA-IB-b	282	267	0.13	0.11	1.3
MTA-O-b	319	305	0.15	0.13	1.5
MTA-M-i	396	375	0.18	0.16	1.5
MTA-E-i	275	251	0.14	0.11	1.5
MTA-B-i	346	327	0.16	0.14	1.5
MTA-IB-i	309	292	0.15	0.13	1.3
MTA-O-i	303	286	0.14	0.12	1.5

Note: S_BET_, S_mic_, V_total_, and V_mic_ are the BET surface areas, microporous surface areas, total pore volumes, and microporous volumes, respectively.

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
