# Peer review of "Facile Synthesis of Super-Microporous Titania–Alumina with Tailored Framework Properties"

_materials, 2020, doi:10.3390/ma13051126_

Round 1
Reviewer 1 Report
Overview:
The authors provide a detailed synthesis and characterization of titanio-alumina super-microsporous materials. The manuscript is generally well-written with a clear synthesis and characterization methodology. The manuscript would benefit from greater clarity as indicated below. While of limited empirical description, the work should be useful to practitioners in the field seeking to achieve super-microporous materials. The following comments are offered to strengthen the quality of the final manuscript.
General Comments:
Lines 31-32: For the sentence, “Mixed titania-alumina oxides exhibit properties superior to those of single-metal oxides (alumina or titania)” it would be beneficial to provide examples and references for those properties that are “superior” and a perspective on what the comparison is based upon.
Lines 39-41: In suggesting a bridging of pore sizes, it would be helpful to describe the benefit bridging length scales provide?
Introduction: Recommend a paragraph on super-microporous materials, including their preparation methods and applications to better put this work in context.
Line 52: Specify vendors for chemicals used.
Line 55: Was the heating done in air?
Line 57: define the acronym/designation “MTA.”
Lines 61-62: Specify the high temperature heating temperatures used. From Figures 4 and 5 they appear to be 550 and 750 °C.
Line 65: Recommend using “Kα” vs “Ka”.
Line 66: Which Quantachrome instrument was used?
Lines 66, 68: Recommend providing references for BET and DFT models.
Line 74: Clarify what “obtained precursor” means. It is not clear from the Methods section which step is referred to.
Line 78: There is an additional small TG-DSC peak at approximately 975°C. What is its significance?
Line 85: The discussion suggests that there are no distinct hysteresis loops; however, many of the isotherms appear to have small hysteresis loops. This should be clarified along with the possible significance of the small hysteresis.
Figure 2: Both panels (a) and (b) do not include the y-axis range from 0-40. Recommend showing the complete data ranges as parts (a) and (b) and move the current panels (a) and (b) to additional panels (c) and (d). In this manner both the complete isotherms with Type I character and the enlarged isotherms may be seen.
Table 1. The parameters SBET, Smic, Vtotal, and Vmic while obvious, are not defined or discussed in the Discussion section. Also, how was Pore Size in the last column determined? The pore size distributions in Figure 3 indicate multiple peaks below 2nm.
Line 120: It is not clear what is meant by “controlling hydrolysis rate” means as there does not appear to be any explicit discussion of hydrolysis rates and supporting data.
Line 121: What would the expected linear relationship with the solvent be, ie what solvent parameter would be correlated with pore size?
Reviewer 2 Report
See attached file

Round 2
Reviewer 2 Report
This version is very much improved and is suitable for publication.